# Severe Pneumonia in Neonates Associated with *Legionella pneumophila*: Case Report and Review of the Literature

**DOI:** 10.3390/pathogens10081031

**Published:** 2021-08-15

**Authors:** Alba Perez Ortiz, Camilla Hahn, Thomas Schaible, Neysan Rafat, Bettina Lange

**Affiliations:** 1Department of Neonatology, University Children’s Hospital Mannheim, University of Heidelberg, Theodor-Kutzer-Ufer 1-3, 68167 Mannheim, Germany; Thomas.Schaible@umm.de (T.S.); Neysan.Rafat@umm.de (N.R.); 2Department of Pediatrics, University Children’s Hospital Mannheim, University of Heidelberg, Theodor-Kutzer-Ufer 1-3, 68167 Mannheim, Germany; Camilla.Hahn@umm.de; 3Department of Hygiene, Medical Faculty Mannheim, University of Heidelberg, Theodor-Kutzer-Ufer 1-3, 68167 Mannheim, Germany; Bettina.Lange@umm.de

**Keywords:** respiratory tract infection, pneumonia, *Legionella*, Legionnaires’ disease, nosocomial infection, neonate, extracorporeal membrane oxygenation

## Abstract

The causative agent of legionellosis is the Gram-negative intracellular bacteria *Legionella* spp. Its clinical presentation varies from a mild febrile illness called Pontiac fever to the severe and possible fatal pneumonia, Legionnaires’ disease. Immunocompromised patients, in particular, are affected. Only a small number of infected neonates are described in the literature. Most of them have been associated with water birth or the use of air humidifiers. In the last five years, a growing number of cases have been reported in Germany by the national institute of disease surveillance and prevention (Robert-Koch Institute). Here, we describe a fatal case report of pulmonary legionellosis with acute respiratory distress syndrome (ARDS), sepsis, associated cutaneous manifestation, and extracorporeal membrane oxygenation in a full-term neonate. Moreover, we present a review of the literature discussing the epidemiology, risk factors, clinical features, diagnostics, treatment options, and prevention for this rare condition in neonates.

## 1. Introduction

Respiratory tract infections in neonates have a significant impact on morbidity and mortality in this fragile patient population. They may be infected in utero, during labor and delivery, or postnatally. The etiology of neonatal pneumonia includes the full spectrum from bacteria and viruses to protozoa. The exposure of the neonate to maternal flora, hospital or domestic environment, or household members play an important role in determining the infecting pathogen. *Legionella* spp. are detected in rare cases as the infecting pathogen in neonatal pneumonia [1]. The incubation period of *Legionella* pneumonia is roughly 2 to 14 days [1]. Most of these infections are of nosocomial origin and associated to contaminated water at various stages of a newborn’s hospitalization, from the delivery room environment, particularly the pool water for water births [2], to sinks [3] or humidifier of the maternity ward [4], to the setting of a neonatal intensive care unit (NICU), predominantly the humidifier of incubators [5]. Early diagnosis and initiation of adequate treatment including extracorporeal membrane oxygenation (ECMO) for refractory respiratory failure are essential for the individual prognosis in neonates, since the mortality rate remains about 50% [1]. We describe a case report of pulmonary legionellosis with acute respiratory distress syndrome (ARDS), sepsis, and associated cutaneous manifestation and ECMO in a full-term neonate. Moreover, we present a review of the literature discussing the epidemiology, risk factors, clinical features, diagnostics, treatment options, and prevention for this rare condition in neonates.

## 2. Case Report

A full-term neonate was born after an uneventful second pregnancy in a hospital birthing-pool elsewhere and discharged 11 h after birth in healthy condition. His parents were consanguine; another family member suffered from an acute lymphatic leukemia and was in treatment.

The neonate presented with fever (38.4 °C) and poor feeding on day 10 of life and was admitted to the hospital with respiratory distress. A neonatal late-onset sepsis (LOS) was assumed and antibiotic treatment with ampicillin and gentamicin was started. Blood and urine cultures were collected and remained sterile over time. The following day the patient’s condition deteriorated, and chest X-ray detected a total pulmonary opacification of the right lung (Figure 1). Respiratory support was necessary due to increasing hypoxemia. Laboratory values showed high C-reactive protein (CRP) level up to 270 mg/L, high Interleukin 6 (IL-6) up to 42.500 pg/mL, and a low white blood cell count (0.5/nL). Two days after admission, the patient required intubation and mechanical ventilation. Antibiotic treatment was escalated based on empirical considerations to meropenem and vancomycin. Additionally, antiviral and antimycotic therapy was initiated.

Due to severe respiratory failure, the patient was transferred to our neonatal ECMO center on day four. Blood tests still revealed high CRP level (max 302 mg/L), low platelet levels (min 107/nL), and a low white blood cell count (minimum 0.70/nL). We continued the antibiotic treatment with meropenem and vancomycin and initiated a surfactant application and treatment with inhalative nitric oxide (NO). Chest X-ray (Figure 1A) and computed tomography (CT) detected bilateral pneumonia with effusions and ground-glass opacities in the right upper lobe. Flexible bronchoscopy was performed to exclude anatomical airway abnormalities. Bronchoalveolar lavage (BAL) samples were analyzed by multiplex polymerase chain reaction (PCR) to screen for viral and atypical bacterial respiratory pathogens.

By optimization of the ventilation strategy (increased PEEP and reducing the driving pressure) the patient’s condition remained stable over two days, but blood parameters for a severe sepsis persisted. Due to parental consanguinity, an inborn immunodeficiency was suggested. Thus, fluorescence-activated cell sorting (FACS) was performed to diagnose primary immunodefiencies, immunoglobulins were applied, and the antibiotic treatment was broadened with trimethoprim-sulfamethoxazol. The following day the patient´s condition deteriorated again with severe hypoxemia on high-frequency oscillatory ventilation (oxygenation index 62). Awaiting further microbiological results, the antibiotic treatment was broadened with amikacin, clarithromycin, and micafungin. FACS analysis for immunodeficiency presented a normal result.

Soon after, on the same day, BAL multiplex PCR results were positive for *Legionella (L.) pneumophila*, while urinary antigen test was negative. Due to *L. pneumophila* pneumonia, the antibiotic treatment was changed to levofloxacin and azithromycin and additionally, venoarterial (VA-) ECMO was initiated. On VA-ECMO, the patient remained stable during the following days. On day 10, the patient developed a generalized erythematous macular rash for 10 days in accordance with cutaneous legionellosis (Figure 1B).

Despite hemodynamic improvement, the pulmonary gas exchange declined and inflammatory markers remained high. At the same time, cultures from tracheal secretions were still positive for *L. pneumophilia* (10^6^/L). ECMO support was discontinued after 22 days, due to thrombotic complications in the ECMO circuit and without significant respiratory improvement. Another CT was performed, which showed a cavitating pneumonia with progressive massive bilateral consolidations (Figure 1C). The patient’s condition deteriorated rapidly, and the child died of cardiopulmonary arrest on day 30 of his severe *Legionella* pneumonia.

Serotyping of the patient’s isolate remained inconclusive. When *L. pneumophila* was detected, water samples for culture were collected by the hygiene team of the hospital elsewhere as well as by health authorities in the private apartment house of the parents. Further molecular–biological workup supported by the German National Reference Center comparing *L. pneumophila* isolate of the patient with those obtained from the maternity bath and the private water distribution system showed no match.

## 3. Material and Methods

A systematic literature review was performed to discuss our own case and to present a survey of the pathogenesis of *Legionella* pneumonia in neonates younger than four weeks of age.

An electronic search of the Medline database (PubMed) of the English and non-English literature from March 1989 through April 2021 using the keywords “*Legionella*, pneumonia and neonate” as well as “Legionnaires’ disease and neonate” was performed. Additionally, we used the keywords “*Legionella*, pneumonia, neonate and ECMO”.

## 4. Results

A total of 191 papers were identified to match the criteria of our search. Of these, 29 fulfilled the research items of “*Legionella*, pneumonia and neonate” (104) as well as “Legionnaires’ disease and neonate” (87) in the title or abstract. The additional search with the keyword “ECMO” identified five reports [6,7,8,9,10,11], all already included in our search. Three papers did not fulfill our search criteria. Most reports were from Europe (n = 16) and North America (N = 10), and only three were from Asia (Table 1). We identified 21 case reports, two articles about one outbreak involving 11 neonates, and six reviews. A total of 45 cases of *Legionella* infection in neonates were analyzed. An overview of all reports found and the relevant clinical information are portrayed in Table 1.

### 4.1. Epidemiology and Environmental Isolation

Of the identified cases, 32 (71%) were nosocomial infections. In most cases, water samples were collected from the hot water system of the hospital environment. *L. pneumophila* was isolated in neonatal respirators [12], oxygen nebulizer [13], incubators, [5] as well as sinks, tank and/or mixer of the maternity ward [3,13,14,15]. In 2013, Yiallouros et al. [4] presented on a large outbreak of *L. pneumophila* pneumonia in term neonates in a private hospital in Cyprus. Microbiological investigations identified a cold-mist humidifier filled with contaminated water from the nursery’s water taps as the source infection [4,16].

The incidence of community-acquired infections was 24% (11 cases), most of them following a water birth at home without careful disinfection [6,7,10,11,17,18,19,20].

In 13 patients (29%), an underlying disease was described, including prematurity [3,5,15,21], immunodeficiency [22,23], congenital heart disease [14,18], and transesophageal fistula [8].

### 4.2. Clinical Findings

In all reports describing the clinical presentation of the disease, acute respiratory distress and radiological signs from pneumonia were present (n = 19) and in nearly all of them (n = 18) leading to severe respiratory insufficiency and mechanical ventilation. Fever was present in 15 patients. Three patients presented with sepsis (Table 1).

### 4.3. Diagnostic Studies

Information about the diagnosis of *Legionella* infection was provided in 27 case reports. *L. pneumophila* was isolated in samples from tracheal aspirate (n = 10), bronchoalveolar lavage (n = 10), or lung tissue obtained at autopsy (n = 7). In most of the cases, the diagnosis was established by culture (n = 24) (Table 1). In three cases, polymerase chain reaction (PCR) provided the diagnosis (Table 1). The use of a urinary antigen test was reported in 15 cases and was positive in 12 patients (80%) (Table 1).

*L. pneumophila* type 1 was the predominant serotype occurring in 22 cases. Serotype 6 was isolated in eight cases and serotype 3 was isolated in the nine cases from the outbreak in Cyprus [4]. The other isolates were serotypes 7 and 8 (Table 1).

### 4.4. Treatment and Outcome

In 27 of the reports, antibiotic treatment was prescribed. Twenty-one patients were treated with macrolide antibiotics; these were empirically prescribed after failure of other antibiotic regimens or after diagnosis of *Legionella*. All patients who did not receive macrolide antibiotics died. In 13 patients, macrolide antibiotics were prescribed in combination with rifampicin (n = 11) or quinolone antibiotics (n = 2) (Table 1).

The use of ECMO was reported in five cases (Table 1). The indication to start extracorporeal support was septic shock with severe respiratory failure in all the patients. Three patients could be successfully decannulated and survived, whereas one patient died. In 30 of the case reports, information on outcome was delivered: 17 patients (57%) recovered (Table 1).

## 5. Discussion

While *Legionella* infection is an established and frequent cause of pneumonia in the adult population, the pediatric literature reveals that *L. pneumophila* is a rare cause of pneumonia in neonates [28,29].

Most of the cases of neonatal legionellosis identified in our review were nosocomial infections following possible aspiration of contaminated hot water during the patients´ hospitalization. Levy and Rubin reviewed, in 1998, nine cases of neonatal Legionnaires´ disease and found that all were hospital acquired infections [26]. Our findings are consistent with those of Greenberg et al. [29]. In their review of pediatric legionellosis, they identified 13 neonatal cases, of which 10 (77%) were nosocomial infections. Immunosuppression and pre-existing lung disease are risk factors for Legionnaires´ disease in both pediatric and adult populations [3,5,15,22,23,29]. In the pediatric patient population reviewed by Greenberg et al., there was an underlying condition in 78% of the cases [29]. In contrast to these results, our review—focusing only on neonatal patients—revealed an underlying disease only in 29% of the patients.

The patient of our case report was, until then, a healthy neonate with severe legionellosis, delivered in a hospital birthing pool, and bathed twice at home in an apartment building. Unfortunately, an environmental link could not be identified since serotyping of the patient’s isolate remained inconclusive.

Clinical features suggesting *Legionella* pneumonia are non-specific (e.g., fever, cough, tachypnea, pneumonia in chest X-ray, acute respiratory distress, and sepsis) [26,29]. The clinical presentation and course of Legionnaires’ disease in neonates is not evident [29] and *Legionella* is not considered a typical causative agent of neonatal pneumonia. For these reasons, its diagnosis requires a high index of suspicion among neonatologists and pediatricians [26,29]. Our patient presented with fever and was admitted to the hospital with progressive respiratory insufficiency requiring mechanical ventilation. Chest X-ray and CT demonstrated bilateral pneumonia. However, the consanguine relationship of the parents and the malignancy of a family member led the neonatologists, in the beginning, to suspect an unknown immunodeficiency syndrome as the foundation for the severe pneumonia and sepsis in the patient, and the diagnosis and treatment of legionellosis were delayed.

Besides respiratory symptoms, Legionnaires’ disease may affect other organs, such as the skin. Nonspecific immune-mediated dermatological manifestations of legionellosis are well known to occur in older patients in the literature [19,32,33,34,35]. Barton et al. [19] were the first to demonstrate a generalized erythematous maculopapular rash over four days in a neonate with *Legionella* pneumonia. Our patient also developed a generalized erythematous maculopapular rash for 10 days on day 10.

When diagnosing legionellosis, the urinary antigen detection is the first-line diagnostic test, although it is limited to the *Legionella pneumophila* serogroup 1 [1]. Combination with the sample culture of the lower respiratory tract is recommended, preferably before starting an antibiotic treatment [1]. The rapid turn-around time and sensitivity of quantitative (q)PCR is advantageous when compared to traditional culture methods [1,36,37]. In our patient, urine antigen testing for *Legionella* was negative. The qPCR and culture of the bronchoalveolar lavage (BAL) samples were positive for *Legionella pneumophila*, but negative for the typical serogroup 1. Therefore, combination diagnostics should be considered, when urine antigen detection is negative, but when an indication for legionellosis is still present.

Positive testing for *Legionella* should immediately lead to adequate antibiotic treatment. Macrolide and quinolone antibiotics are suggested as first-line therapy of severe *Legionella* pneumonia. The recommended duration of azithromycin treatment is 3–5 days and for levofloxacin, 5–10 days [1]. In our patient, antibiotic treatment with ß-lactam antibiotic and aminoglycoside was changed to meropenem and vancomycin, both ineffective against *L. pneumophila*. When the diagnosis of Legionnaires´ disease was established, adequate therapy with levofloxacin and azithromycin was started and improvement of sepsis was observed. Thus, in any case of pneumonia unresponsive to antibiotics, Legionnaires’ disease should be considered and specific tests to verify the diagnosis should be performed [29].

*Legionella* spp. is a rare cause of neonatal pneumonia, but the high mortality rate of 50% in children younger than one year is still being discussed in the literature [29]. Three cases of successful application of ECMO in neonates affected with severe Legionnaires´ disease are reported in the literature [8,20,23]. Dorfmann et al. [20] recommend the use of ECLS early in the course of infection to improve outcomes in this subset of patients. The Extracorporeal Life Support Organization (ELSO) has defined indications, contraindications, and modes of support for neonatal ECMO [38]. Neonates with severe respiratory and/or cardiac failure with a high likelihood of mortality and potentially reversible etiology are eligible for ECMO [38]. The indications include: 1) inadequate tissue oxygen delivery despite maximal therapy (rising lactate, worsening metabolic acidosis, signs of end organ dysfunction), 2) severe hypoxic respiratory failure with acute decompensation (PaO_2_ < 40 mmHg), 3) oxygenation index with sustained elevation and no improvement, and 4) severe pulmonary hypertension with evidence of right ventricular dysfunction and/or left ventricular dysfunction [38]. Contraindications include lethal chromosomal disorders or other lethal anomalies, severe brain damage, uncontrollable bleeding, and significant intraventricular hemorrhage [38]. Venoarterial (VA) ECMO provides cardiac and respiratory support, while venovenous (VV) ECMO does not offer hemodynamic support and is preferred for respiratory support only [38]. In our case, VA-ECMO was initiated on day eight of fulminant sepsis and severe pneumonia, when the diagnosis of Legionnaires’ disease was made. Antibiotic treatment was adjusted on the same day and improvement of the patient´s condition was observed. Unfortunately, the neonate died after 22 days, still suffering from respiratory failure due to destroyed lung parenchyma. Post-mortem examination was declined by the parents.

Since water is the major natural reservoir for *Legionellae*, and the bacteria are found worldwide in many different natural and artificial aquatic environments, prophylactic measures have to be established. The water safety plan of the WHO is an instrument to maintain preventive risk management systems to control *Legionella* spp. [39]. Surveillance of water-related infectious diseases associated with water-supply systems is recommended for health-care facilities as well as private households. Routine culturing of water samples is regulated in hospitals by law, but the private use of heated birthing-pools is still without any control [6]. Review of the literature demonstrates for Europe and North America the ability to put into practice surveillance and preventive programs for Legionnaires’ disease. Thus, routine culturing of water samples in any birthing-pool may protect neonates in the future. Analysis of our own case report attested regular check of the water supply system of the birthing-pool in the maternity ward and installation of point of use filters in the birthing-pool of the hospital.

## 6. Conclusions

In conclusion, *Legionella* pneumonia is a life-threatening infectious disease in neonates. Physicians may suspect *Legionella pneumophila* in any case of pneumonia and/or sepsis, non-responsive to ß-lactam antibiotic therapy. Clinical features are variable; extrapulmonary manifestation such as cutaneous legionellosis is possible. The diagnostic gold standard is urinary antigen testing as well as culture and qPCR of tracheobronchial samples. First-line therapy is based on fluorochinolone and macrolide. Early use of ECMO should be considered in case of progressive respiratory failure. International and national water safety plans have to maintain surveillance and infection prevention to avoid severe *Legionella* pneumonia in neonates.

## Figures and Tables

**Figure 1 pathogens-10-01031-f001:**
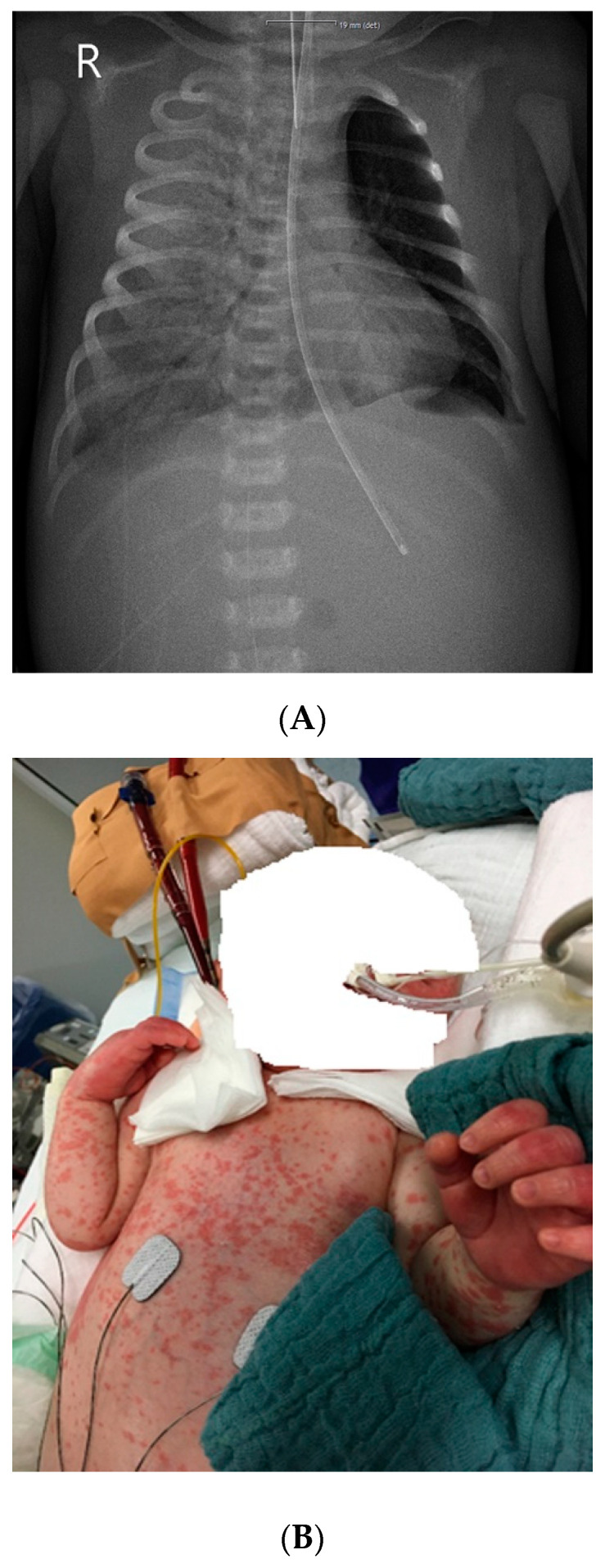
Imaging results and clinical findings in a 14-day-old boy after a water birth. (**A**) Chest X-ray shows a total pulmonary opacification of the right lung (4 days after hospitalization). (**B**) Cutaneous legionellosis manifested itself during venoarterial extracorporeal membrane oxygenation for severe respiratory failure (10 days after hospitalization). (**C**) Chest computed tomography scan showing a cavitating pneumonia with progressive massive bilateral consolidations after discontinuation of venoarterial extracorporeal membrane oxygenation.

**Table 1 pathogens-10-01031-t001:** *Legionella pneumophila* pneumonia in neonates: review of the literature, 1989–2021.

Reference, Country	Type of Publication	No. of Cases	Age at Diagnosis (d)	Positive Diagnostics	Source of *L. pneumophila*	Serotype	Treatment with Macrolide Antibiotics	ECMO	Outcome
Marcelles AF et al., 1989, [12] Spain–Europe	CR	1	1	Postmortem autopsy	^1^ Humidifier of the respirator	6	No	No	Death
Aubert G et al., 1990, [13] France–Europe	CR	1	10	BAL culture	^1^ Oxygen nebulizer, feeding bottle system	1,8	Unknown	No	Recovery
Greene KA et al., 1990, [14] North America	CR	* 1	12	Postmortem autopsy	^1^ Not detected	1	No	No	Death
Ferrer A et al., 1990, [22] Spain–Europe	CR (Review)	* 4	10–2555	Pleural fluid, tracheal aspirate, BAL culture, postmortem biopsy	^1^ Not detected	6	Yes	No	Recovery/ 3× death
Horie H et al., 1992, [24] Japan–Asia	CR	1	5	Postmortem autopsy	^1^ Unknown	1	No	No	Death
Womack SJ et al., 1992, [21] North America	CR	* 1	16	Postmortem Autopsy	Respirator	1	No	No	Death
Ahrens F et al., 1993, [25] Germany–Europe	CR	1	5	BAL culture	^1^ Environmental cultures	1	Yes	No	Recovery
Holmberg Jr RE et al., 1993, [15] North America	CR	* 1	31	Tracheal aspirate culture	^1^ Nursery sink, nurse’s wet hand	6	Yes (+rifampicin)	No	Recovery
Lück PC et al., 1994, [5] Germany–Europe	CR	* 1	10	Tracheal aspirate culture	^1^ Humidifier of the respirator	1	No	No	Death
Levy I and Rubin LG, 1998, [26] North America	Review	* 9	4–31	Pleural fluid, tracheal aspirate, BAL culture, DFA assay, postmortem biopsy	^1^ Humidifier, nebulizer, incubator	1,6	Yes (+rifampicin)	No	Recovery/ death
Franzin et al., 2001, [2] Italy–Europe	CR	1	7	Urinary antigen test, DFA assay	^1^ Pool water (water birth)	1	Yes	No	Recovery
Skogberg K et al., 2002, [27] Finland–Europe	CR	1	7	Urinary antigen test; BAL culture, DFA assay	^2^ Apartment building	6	Yes (+rifampicin)	No	Recovery
Nagai T et al., 2003, [17] Japan–Asia	CR	1	4	Postmortem autopsy (PCR)	^2^ Bath tub (water birth)	1,6	None	No	Death
Franzin L et al., 2004, [28] Italy–Europe	(CR) Review	* 11	7	Urinary antigen test, DFA assay	^1^ Pool water (water birth)	1	Yes	No	Recovery
Greenberg D et al., 2006, [29] North America	Review	* 76	>5	Culture methodology, DFA assay	^1^ Environmental cultures	1	Yes	No	Recovery/ death
Eurosurveillance, 2009, [16] Europe	Outbreak	11	6–12	Urinary antigen test, BAL	^1^ Humidifier	1,3	Yes (+rifampicin)	No	Recovery/ death
Shachor-Meyouhas Y et al., 2010, [3] Israel–Asia	CR	* 1	11	Postmortem Autopsy (culture)	^1^ Sink of the ward	1	Yes	No	Death
Yu VL and Lee TC, 2010, [30] North America	Review		4–11	Urinary antigen test, culture	^1^ Environmental cultures	1,6	Yes	No	Unknown
Teare L, Millership S, 2012, [31] UK–Europe	Review				Pool water (water birth)	1	Not detected	No	Unknown
Yiallouros PK et al., 2013, [4] Cyprus–Europe	Outbreak	9	6–12	Urinary antigen test, tracheal aspirate culture	^1^ Humidifier	1,3	Yes (+rifampicin)	No	Recovery/ 3x death
Phin N et al., 2014, [6] UK–Europe	CR	1	3	BAL culture	^2^ Birthing pool (water birth)	1	ECMO	Yes	Unknown
Fritschel E et al., 2015, [7] North America	CR	1	6	Urinary antigen test, tracheal aspirate PCR	^2^ Collapsible tub (water birth)	1	No	Yes	Death
Moscatelli A et al., 2015, [8] Italy–Europe	CR	* 1	12	BAL PCR	^1^ Environmental cultures	Unknown	Yes (+rifampicin)	Yes	Recovery
Fremgen L, 2015, [10] North America	CR	1			^2^ Pool water (water birth)	Not detected	Not detected	No	Unknown
Collins SL et al., 2016, [11] UK–Europe	CR	1	3	BAL culture/ PCR	^2^ Birthing pool (water birth)	1	Not detected	No	Unknown
Granseth G et al., 2016, [18] North America	CR	* 2	1,3	Urinary antigen test, tracheal aspirate, BAL culture	^2^ Tub (water birth)	1,6	Yes	No	Recovery
Barton M et al., 2017, [19] UK–Europe	CR	1	8	Urinary antigen test, tracheal aspirate culture	^2^ Hot tub (water birth)	6	Yes (+rifampicin)	No	Recovery
Leruste A et al., 2017, [23] France-Europe	CR	* 1	27	Tracheal aspirate, BAL culture/PCR	^2^ Not detected	3	Yes	Yes	Recovery
Dorfman MV et al., 2020, [20] North America	Review	1	6	Urinary antigen test, tracheal aspirate culture	^2^ Water birth	Unknown	Yes	Yes	Recovery

No., number; d, days; CR, case report; L, Legionella; ECMO, extracorporeal membrane oxygenation; BAL, bronchoalveolar lavage; DFA, direct fluorescent antibody; PCR, polymerase chain reaction; UK, United Kingdom. * Underlying disease (prematurity, immunodeficiency; congenital heart disease, transesophageal fistula); ^1^ Nosocomial; ^2^ community-acquired.

## Data Availability

Data sharing not applicable.

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
