# Peer review of "Severe Pneumonia in Neonates Associated with Legionella pneumophila: Case Report and Review of the Literature"

_pathogens, 2021, doi:10.3390/pathogens10081031_

Round 1

Author Response

1) “I think that will be good that the authors should suggest more prophylaxis and immunostimulation, because Legionella spp. is potentially everywhere. Very important is that before treatment should make microbiological diagnosis with antibiotic pattern.” 

We thank the reviewer for reviewing our manuscript and her/his thoughtful comments.

In case of prophylaxis, we focused in the discussion section on the safety plan of the WHO and the surveillance of water-related infectious diseases associated with water supply systems for health-care facilities as well as private households:  

Since water is the major natural reservoir for Legionellae, and the bacteria are found worldwide in many different natural and artificial aquatic environments, prophylactic measures have to be established. The water safety plan of the WHO is an instrument to maintain preventive risk management systems to control Legionella spp. [39]. Surveillance of water-related infectious diseases associated with water-supply systems is recommended for health-care facilities as well as private households. Routine culturing of water samples is regulated in hospitals by law, but the private use of heated birthing-pools is still without any control [6]. Review of the literature demonstrates for Europe and North America the ability to put into practice surveillance and preventive programs for Legionnaires’ disease. Thus, routine culturing of water samples in any birthing-pool may protect neonates in the future. Analysis of our own case report attested regular check of the water supply system of the birthing-pool in the maternity ward and installation of point of use filters in the birthing-pool of the hospital.“

(lines 269-281 in the revised manuscript)  

We agree that before starting antibiotic treatment, microbiological diagnostic testing is important for the outcome of the patient and mentioned this in the manuscript: 

Combination with sample culture of the lower respiratory tract is recommended, preferable before starting antibiotic treatment [1]. The rapid turn-around time and sensitivity of quantitative (q)PCR is advantageous when compared to traditional culture methods [1,36,37].“

(lines 229-232 in the revised manuscript)

Reviewer 2 Report

Really interesting paper covering a very important issue. Well written, few minor comments below

Abstract - Legionella needs to be italicised (check throughout document)

Line 34 - Legionella needs to be italicised - rather than Legionella pneumonia consider using Legionnaires disease?

lines 97-100 How was the molecularbiological workup comparing the clinical isolate to the bath and private water samples conducted? Could you please provide some more information. Was culture used to sample the water? Were biofilm swabs taken from the bath? The lack of a match could be due to limitations with environmental sampling. 

Methods - I'm concerned that by not including the search term "Legionnaires disease and neonate" you may have missed some relevant articles. Could you please check and add any additional article if required

lines 254- 256- In this sentence are you saying this was currently implemented at the hospital associated with the case study? If so could you add some details around this to the case study section? A bit more information here around the environmental side of the case study would be very interesting.

Otherwise if this is a recommendation (great idea!) then please reword to make this clear - the evidence support the use of these control measures to protect neonates. 

Author Response

“Really interesting paper covering a very important issue. Well written, few minor comments below.” 

We thank the reviewer for reviewing our manuscript and her/his thoughtful comments.

1.) “Abstract – Legionella needs to be italicised (check throughout document).” 

We have now italicised “Legionella” in the abstract and corrected some more mistakes, all marked in the manuscript.   

2.) “Line 34 – Legionella needs to be italicised – rather than Legionella pneumonia consider using Legionnaires disease?” 

We have now italicised “Legionella” throughout the entire manuscript and corrected "Legionella pneumonia" into "Legionnaires disease" .   

3.) “lines 97-100 How was the molecularbiological workup comparing the clinical isolate to the bath and private water samples conducted? Could you please provide some more information. Was culture used to sample the water? Were biofilm swabs taken from the bath? The lack of a match could be due to limitations with environmental sampling” 

In the revised manuscript we have now mentioned the molecular-biological workup:  

Serotyping of the boy’s isolate remained inconclusive. When L. pneumophila was detected, water samples for culture were collected by the hygiene team of the hospital elsewhere as well as by health authorities in the private apartment house of the parents. Further molecular-biological workup supported by the German National Reference Center comparing L. pneumophila isolate of the patient with those obtained from the maternity bath and the private water distribution system showed no match.”

(lines 103-108 in the revised manuscript)  

Water samples were collected by the hygiene team of the hospital elsewhere after removal of point-of-use filters, often used in German birthing-pools to protect the neonate during delivery. To the best of our knowledge, biofilm swabs revealed no Legionella. We agree with the reviewer that the lack of match could be due to the limitations with environmental sampling conducted in a hospital and private household elsewhere.

4.) Methods - I'm concerned that by not including the search term "Legionnaires disease and neonate" you may have missed some relevant articles. Could you please check and add any additional article if required”

We have performed now an additional electronic search of the Medline database (PubMed) including the search term “Legionnaires`disease and neonate”.

A total of 84 papers were identified, 29 fulfilled the research items in the title or abstract, 26 were already included in our review.

The following three out of these 29 papers did not fulfil our research criteria:

1.) Szewc AM, Taylor S, Cage GD, de Mello D. Male neonate with Legionellosis. Lab Med 2017;49(1): e9-e13.doi:10.1093/labmed/lmx073

The authors report on a 7-month-old boy, which did not fulfil our criteria of neonates (case report).

2.) Rayet I and Teyssier G. Nosocomial legionellosis in newborn infants. Arch Fr Pediatr 1990;47(6):469

The article is in French without any accurate translation in English (case report).

3.) Kodousek R, Dusek J, Zidova L, Macak J, Volenjnik J. An unusual case of fatal macrophagic pneumonia in a 12-days old infant, caused by a gram-negative Legionella-like microorganism. Acta Univ OPalacki Olomuc Fa Med 1982;102:131-4

In this case report the authors only report on a “Legionella-like microorganism”.

Therefore, in our opinion none of these articles must be added.

We have revised our search items in the “Material and Methods” section:

“An electronic search of Medline database (PubMed) of English and non-English literature (with translation) from March 1989 throughout April 2021 using the keywords “Legionella, pneumonia and neonate” as well as “Legionnaires’ disease and neonate” was performed. Additionally, we used the keywords “Legionella, pneumonia, neonate and ECMO”.

(lines 122-125 in the revised manuscript)

We have added these three additional papers in the “Results” section and mentioned that these articles did not fulfil our search criteria:

„A total of 191 papers were identified to match the criteria of our search. Of these, 29 fulfilled the research items of “Legionella, pneumonia and neonate” (104) as well as “Legionnaires’ disease and neonate” (87) in the title or abstract. The additional search with the keyword “ECMO” identified five reports [6–11], all already included in our search. Three papers did not fulfill our search criteria.“

 (lines 129-133 in the revised manuscript)

We hope we have responded adequately to the reviewers comments.

5.) lines 254- 256- In this sentence are you saying this was currently implemented at the hospital associated with the case study? If so could you add some details around this to the case study section? A bit more information here around the environmental side of the case study would be very interesting.

Otherwise if this is a recommendation (great idea!) then please reword to make this clear - the evidence support the use of these control measures to protect neonates.“ 

As the reviewer has correctly pointed out, this was meant as a recommendation. We recommend routine culturing of water samples in a hospital as well as in the private setting to protect any neonate in a birthing-pool.

Therefore, we have added this statement in the revised version of the manuscript to highlight this.

Thus, routine culturing of water samples in any birthing-pool may protect neonates in the future”

(lines 277-279 in the revised manuscript)